# Gender Differences in All-Cause Mortality after Acute Myocardial Infarction: Evidence for a Gender–Age Interaction

**DOI:** 10.3390/jcm11030541

**Published:** 2022-01-21

**Authors:** Pil Sang Song, Mi Joo Kim, Seok-Woo Seong, Si Wan Choi, Hyeon-Cheol Gwon, Seung-Ho Hur, Seung-Woon Rha, Chang-Hwan Yoon, Myung Ho Jeong, Jin-Ok Jeong

**Affiliations:** 1Division of Cardiology, Department of Internal Medicine, Chungnam National University Hospital, Chungnam National University College of Medicine, Daejeon 305764, Korea; hjmj48600@hanmail.net (M.J.K.); outschool@naver.com (S.-W.S.); siwanc@cnu.ac.kr (S.W.C.); 2Division of Cardiology, Department of Medicine, Heart Vascular Stroke Institute, Samsung Medical Center, Sungkyunkwan University School of Medicine, Seoul 06351, Korea; hcgwon@naver.com; 3Cardiovascular Medicine, Keimyung University Dongsan Medical Center, Deagu 41932, Korea; shur@dsmc.or.kr; 4Cardiovascular Center, Korea University Guro Hospital, Seoul 08308, Korea; swrha617@yahoo.co.kr; 5Cardiovascular Center, Seoul National University Bundang Hospital, Seongnam 13620, Korea; changhwanyoon@gmail.com; 6Chonnam National University Hospital, Gwangju 61469, Korea; myungho@chollian.net

**Keywords:** acute myocardial infarction, gender differences, interaction term of gender with age

## Abstract

Gender difference studies in mortality after acute myocardial infarction (AMI) have shown inconsistent results. A total of 13,104 patients from the KAMIR-NIH between November 2011 and December 2015 were classified into young (*n* = 3837 [29.3%]) and elderly (*n* = 9267 [70.7%]) patients. For the study, women <65 and men <55 years of age were considered “young”. In the adjusted model of the entire cohort, there was no significant difference in three-year all-cause mortality between women and men (17.8% vs. 10.3%; adjusted hazard ratio [HR], 0.953; 95% confidence interval [CI], 0.799–1.137). However, when the entire cohort was subdivided into two age groups, young women showed an 84.3% higher mortality rate than young men (adjusted HR, 1.843; 95% CI, 1.098–3.095). Contrariwise, elderly women patients had a 20.4% lower hazard of mortality compared with elderly men (adjusted HR, 0.796; 95% CI, 0.682–0.929). The interaction of gender with age was significant, even after multiple adjustments (adjusted *p* for interaction = 0.003). The purpose of this study was to assess whether gender differences depend on the patients’ age. Based on our analysis, higher mortality of young women remains even in the contemporary era of AMI. A better understanding of the mechanisms underlying these differences is warranted.

## 1. Introduction

Previous studies have consistently reported that female patients with acute myocardial infarction (AMI) have higher unadjusted mortality rates than men, especially for short-term follow-up [1,2]. Women present with AMI at a later age, with greater cardiovascular risk burdens, and with less opportunity to receive guideline-recommended therapies. However, data are conflicting regarding whether mortality remains higher in women after adjusting for differences in age and other prognostic factors [3,4]. Differences in the age distribution of study samples might explain the inconsistency in results across studies. Although gender differences have been well documented in older patients after AMI [5], uncertainty arises over whether this trend extends to younger patients even in the contemporary era of AMI. In addition, few studies have investigated whether the association between gender and long-term mortality changes according to age. Therefore, the purpose of this study was to determine how patient gender and age influence the management and clinical outcomes of AMI, irrespective of differences in comorbidity.

## 2. Methods

### 2.1. Study Population and Data Management

The data were obtained from the database of the Korea Acute Myocardial Infarction Registry–National Institutes of Health (KAMIR-NIH). The KAMIR-NIH is a prospective, multi-centre, observational cohort study for patients with AMI registered at 20 major cardiovascular centres in Korea from November 2011 to December 2015 [6]. Trained clinical research coordinators collected all data using a web-based case report form in the Internet-based Clinical Research and Trial management system (iCReaT), a data management system established by the Centres for Disease Control and Prevention, Ministry of Health and Welfare, Korea (iCReaT study no. C110016). All events were identified by a physician and confirmed by the principal investigator of each hospital. This study was conducted in accordance with the Declaration of Helsinki and was approved by the institutional review board of each participating institution. Written informed consent was obtained from all patients. The protocol was approved by the ethics committee of each participating institution.

In most analyses of the study, patients were classified into one of two age groups, young or elderly. For the current analysis, we defined “young” as <65 years of age in women and as <55 years of age in men. This age cut-off point was chosen because women are often older when they present with their first AMI, at an average age of 71.8 years compared with 65 years for men [7]. In addition, the age threshold for what is considered “young” based on multiple guidelines from a family history of young coronary artery disease studies is defined as males <55 years old or females <65 years old [7,8].

### 2.2. Patient Treatment

Percutaneous coronary intervention (PCI) was performed according to standard guidelines. Prior to PCI, all patients received a loading dose of aspirin (300 mg) and a P2Y 12 inhibitor (ticagrelor 180 mg; prasugrel 60 mg; or clopidogrel 300 to 600 mg). Route selection for catheterisation, adjunctive drugs to support PCI, and use of thrombus aspiration or intravascular imaging were left to the operator’s discretion. After PCI, the patients received lifelong aspirin plus a P2Y12 inhibitor for >1 year unless there was an unavoidable reason for antiplatelet agent discontinuation. Medications such as renin–angiotensin–aldosterone system (RAAS) blockers, beta-blockers, and statins were prescribed as per the guidelines.

### 2.3. Study End Points and Definitions

The primary endpoint of the study was 3-year all-cause mortality; the secondary outcomes were in-hospital all-cause mortality, recurrent myocardial infarction, re-hospitalisation for heart failure, and major adverse cardiac events (MACE) (a composite of all-cause mortality, recurrent myocardial infarction, and re-hospitalisation for heart failure) at three years. All deaths were considered cardiac unless an undisputed non-cardiac cause was present.

The diagnosis of AMI was based on the detection of an increase and/or decrease of cardiac biomarkers (creatine kinase-myocardial band and troponin I or T), with at least one value above the 99th percentile upper reference limit. Additionally, at least one of these was required to be present: symptoms of ischaemia, new ischaemic electrocardiography changes, development of pathological Q waves, imaging evidence of new loss of viable myocardium, or new regional wall motion abnormality in a pattern consistent with an ischaemic etiology [9,10]. For the current analysis, we defined “myocardial infarction with non-occlusive coronary artery (MINOCA)” only as an AMI without obstructive disease on angiography (i.e., no coronary artery stenosis >50%) in any major epicardial vessel. Given that the clinical and angiographic variables evaluated in the present analysis were collected at a time when the MINOCA definition was not yet available, the current definitions could partly differ from the original ones provided by the guidelines of the major society of cardiology [11].

### 2.4. Statistical Analysis

Categorical variables are presented as number of cases and percentages and were compared using the chi-square test or Fisher exact test. Continuous variables are expressed as mean ± SD or median (interquartile range [IQR]) and were compared using one-way analysis of variance or Kruskal–Wallis test, as appropriate. The chronological trend of the clinical outcomes is presented as a Kaplan–Meier survival curve and compared according to gender.

The Log-rank test was performed for comparison of the differences in clinical outcomes. We constructed a series of hierarchical Cox proportional hazard models to calculate hazard ratio (HR) and 95% confidence interval (CI) for all clinical events using a backward stepwise method in a sequential fashion. The first model included gender as the sole explanatory variable (univariate analysis). The second model included gender and age (as a continuous variable). In the third model, other characteristics were added. These added characteristics were baseline variables with *p* < 0.05 in the univariate analysis and any other baseline variables judged to be of clinical relevance from the previously published literature: age categorised into 2 age groups (young versus elderly); gender; body mass index; no chest pain at presentation; Killip class; systolic blood pressure; heart rate; current smoker status; diabetes mellitus; hypertension; dyslipidemia; previous myocardial infarction; previous cerebrovascular accident; haemoglobin; white blood cell count; creatinine clearance; left ventricular ejection fraction; coronary angiography; use of glycoprotein IIb/IIIa inhibitors; multivessel disease; anterior AMI; post-PCI TIMI (thrombolysis in myocardial infarction) 2 or 3 flow; major bleeding or acute kidney injury during hospitalisation; mechanical cardiac support; and use of beta-blockers, RAAS blockers, potent P2Y12 inhibitors (ticagrelor or prasugrel), or oral anticoagulants at discharge. This method of hierarchical modeling allowed for assessment of the impact of each of the features sequentially added to the model on the association between gender and mortality.

Next, with further statistical analysis, the interaction term of gender with age was added to the previous third Cox model. This interaction term tested the hypothesis that the association between gender and mortality differs between men and women according to 2 age groups even after adjusting for a number of differences in comorbidity and other risk factors. This term also allowed the calculation of HRs of mortality for women versus men within each age group. In addition, to be reassured that the results were not dependent on the age cut-off point chosen, we repeated the analyses using gender–age interactions that treated age in different ways, i.e., age as a continuous variable. The proportionality assumption was assessed by log-minus-log plots. Two-sided *p* values <0.05 were considered statistically significant. The statistical tests were performed using IBM SPSS version 23 (SPSS Inc., Chicago, IL, USA) and R programming version 3.6.1 (The R Foundation for Statistical Computing, Vienna, Austria).

## 3. Results

### 3.1. Baseline Characteristics

Among 13,104 patients enrolled in KAMIR-NIH, the young group comprised 3837 (29.3%) patients, of which 746 were women and 3091 were men. The elderly group comprised 9267 (70.7%) patients, of which 2672 were women and 6595 were men (Appendix A). Table 1 shows comparisons of baseline characteristics between women and men after stratification by age. In both age groups, women were more likely to report a history of diabetes, hypertension, or stroke and have Killip class ≥ 2, renal insufficiency, or anaemia on admission. Women were less likely to smoke. These gender differences were larger in young patients with AMI (interaction effect *p* < 0.05).

### 3.2. In-Hospital Characteristics

During AMI hospitalisation, women were less likely to present with ST-segment elevation AMI and were more likely to experience the MINOCA (Table 2). In both age brackets, women were less likely to receive glycoprotein IIb/IIIa inhibitors, PCI with or without coronary stenting, and statin than men. These differences between men and women were more pronounced in the young AMI group (interaction effect *p* < 0.05). In addition, lower use of P2Y12 inhibitors, β-blockers, or RAAS blockers in women with respect to men was noted only in the younger age group (interaction effect *p* < 0.05). Though there were no gender differences in major bleeding and mechanical cardiac support during hospitalisation, significantly more women (*n* = 182; 5.3%) than men (*n* = 322; 3.3%) died during hospitalisation (adjusted HR, 1.752, 95% CI, 1.060–2.895; *p* = 0.029) (Table 3). Especially, in-hospital all-cause mortality was more prevalent among women than men in the young AMI group (2.7% vs. 1.2%; adjusted HR, 3.305, 95% CI, 1.211–9.019, *p* = 0.020; adjusted *p* value for the interaction = 0.020) (Table 4).

### 3.3. Three-Year Clinical Outcomes

Kaplan–Meier curves for all-cause mortality in all study patients are shown in Figure 1. In the entire sample, the unadjusted incidence of three-year all-cause mortality was significantly higher in women than men (17.8% vs. 10.3%, unadjusted HR, 1.798, 95% CI, 1.626–1.989, *p* < 0.001). Simple age adjustment attenuated the differences with regard to mortality between women and men: women had a lower all-cause mortality rate (age-adjusted HR, 0.877, 95% CI, 0.788–0.975, *p* = 0.016). After additional adjustment for comorbidity, clinical severity variables, and process of care, however, there were no significant differences in all-cause mortality between women and men in the entire cohort (adjusted HR, 0.953, 95% CI, 0.799–1.137, *p* = 0.596, Table 3).

The second part of the analysis assessed the role of the gender–age interaction between men and women and provided separate estimates of the association between gender and all-cause mortality in the two age groups after adjusting for all the factors mentioned earlier. When the gender–age interaction was added to the model, the *p* value for the interaction between gender and age was 0.003 in the multivariate adjusted model (Table 4). Hazard ratios calculated from this model indicated that the risks of all-cause mortality were almost 84% higher in women compared with men in the young age group (adjusted HR, 1.843; 95% CI, 1.098–3.095, *p* = 0.021, Figure 2A). However, the same model showed that women had almost 20% lower hazard of mortality compared with men in the elderly group (adjusted HR, 0.796; 95% CI, 0.682–0.929; *p* = 0.004, Figure 2B) (Table 4). When we used a different scale of measurement for age (as a continuous variable), similar results were indicated, with a *p* value of 0.005 for the interaction between gender and age in the multivariate adjusted model for all-cause mortality.

## 4. Discussion

The present study demonstrated that (1) the proportion of younger patients with AMI was relatively high, representing >1 of 4 patients of the cohort (29.3%), (2) baseline and treatment characteristics differed between women and men, especially in young patients with AMI. Women had a higher prevalence of comorbidities such as diabetes, hypertension, and previous cerebrovascular accident but were less likely to receive PCI or evidence-based pharmacologic therapies, and (3) after multiple adjustments, the risk of mortality from any-cause was similar between women and men during the three-year follow-up. However, interaction between gender and age have a significant impact on long-term clinical outcomes; the 3-year all-cause mortality for women compared with men of similar age increased as age decreased after multiple adjustments (Appendix A).

In the present study, women had higher unadjusted mortality rates during three-year follow-ups. Consistent with previous observations [12], however, age alone appears to account for gender differences in all-cause mortality in the overall population, reducing the hazard ratio for women compared with men from 1.798 to 0.877 in the entire population of our registry. As can be inferred from these results, differences in the age distribution of the study population may explain some of the discrepancies found in previous studies; the increased risk of long-term mortality in young or elderly women can be masked by integrated testing of all age groups. To date, however, little is known about the interaction between gender and age on long-term clinical outcomes [13]. In the current study, including interactions between gender and age in a multiple adjustment model yielded different end results. When comparing the mortality experiences of women and men within age groups by examining the interaction between gender and age, our findings showed that the mortality rate of women increased compared to that of men as age decreased; younger women had a higher 3-year mortality rate than men of similar age. The same was true even after adjustment for underlying comorbidities, clinical severity at the time of admission, in-hospital treatments and complications, and managements during follow-up. Among elderly patients with AMI, however, the inverse appeared to be true; the mortality rate was lower in elderly women than in men.

Why do women after AMI experience worse clinical outcomes at a younger age, but not as they get older? Probably, the age-dependent impact of gender on long-term mortality after AMI can be explained by age-related gender differences in underlying comorbidities and treatment processes (e.g., no chest pain at presentation, more complex baseline risk profiles, absence of invasive therapy [14], and less-than-optimal secondary prevention). First, we found a higher proportion of underlying comorbidities and risk factors in women compared to men. In particular, these adverse cardiovascular risk characteristics were more pronounced in young than in elderly age groups. Due to the protective effect of estrogen in premenopausal women, women are relatively free from coronary atherosclerosis until about 75 years of age [15]. Young women who develop AMI may be predisposed to the disease because of its early onset, high-severity risk factors for coronary artery disease, and/or some unknown risk factors [16]. For these same reasons, the prognosis after AMI can be aggressive in these women. Second, the use of evidence-proven therapies for AMI, such as invasive therapy, use of β-blockers, use of RAAS blockers, and use of statin, is lower for women, especially in the younger age group. Gurwitz et al. presented data that women under 55 years of age had the lowest use of thrombolytics and β-blockers compared to men of similar age. [17]. However, there has been little investigation into whether gender differences in AMI treatment differ with age in the contemporary era. We became aware of the lower use of such evidence-proven therapeutic interventions in women compared to men. The difference was more pronounced in the younger age group. This may be due to the bias of clinicians in evaluating and treating women, especially young women [18]. These findings provide clinically relevant information to physicians, and efforts are needed to ensure equality in caring for women and men after AMI.

In recent years, there has been an overall decrease in cardiovascular disease prevalence and AMI mortality in the general population [19]. Despite improvements in clinical outcomes, however, the rates of hospitalisation for AMI in younger women have increased over the past 20 years. Cardiovascular mortality has also been higher for young women [20]. These trends are thought to be caused by gender differences in baseline cardiovascular risk profiles and guideline-recommended therapies even in the contemporary era of AMI. For improving outcomes in young women with AMI, therefore, there is a need to advance more effective awareness of their distinct cardiovascular risk profile and to develop more comprehensive treatment strategies specific to this population. As an example, expanding initiatives such as the American Heart Association Go Red for Women campaign [21] to increase awareness of cardiovascular disease risk in women through media and other outlets should be encouraged.

### Study Limitations

Our study had several limitations. The KAMIR-NIH is an observational study, and our results might be limited by various biases and unmeasured or inadequately measured potential confounders. The association between a patient’s gender and clinical outcome may depend on confounding factors, including lifestyle and behavioral factors. We are aware that other unmeasured confounding factors may remain while adjusting for a wide range of baseline covariates including demographics, underlying comorbidities, laboratory values, and in-hospital treatments. For example, there were no data on patient preferences or psychosocial factors that were shown to influence patient outcomes [22,23]. There are also no adequate data on the long-term maintenance of prescribed medications. Follow-up data were collected prospectively, and cause of death was typically determined through prospective scheduled surveillance in the KAMIR-NIH registry. Despite this methodology, however, it is possible that misclassification of causes of death occurred in some. More so, the risk of misclassification is expected to be greater in populations with multiple competing comorbidities, such as women in the young group of our study cohort. There were fewer women with AMI than men in our cohort. Therefore, the ability of the study to assess differences in mortality rates was limited. In addition, the sample size may be inappropriate to evaluate gender differences within each age group. Therefore, a larger sample is needed to conclude that women have a much higher mortality rate than men at a younger age. Our artificial definition of different age cut-offs between genders can be quite arbitrary, especially when examining a gender–age interaction. Information on the MINOCA was imperfect in the KAMIR-NIH registry. Specifically, we did not have data on some aetiologies of MINOCA, such as spontaneous coronary artery dissection [24,25]. Finally, the results of our analysis should be tested in other validation cohorts, especially Western patients.

## 5. Conclusions

Even in the contemporary era, young women who experience AMI are more burdened with traditional risk factors than young men. Young women are less likely to be treated with invasive procedures and evidence-proven secondary preventive medical therapies, resulting in a significantly higher rate of long-term all-cause mortality after AMI. Further studies need to address the mechanisms underlying these gender differences and the poor prognosis of young women with AMI.

## Figures and Tables

**Figure 1 jcm-11-00541-f001:**
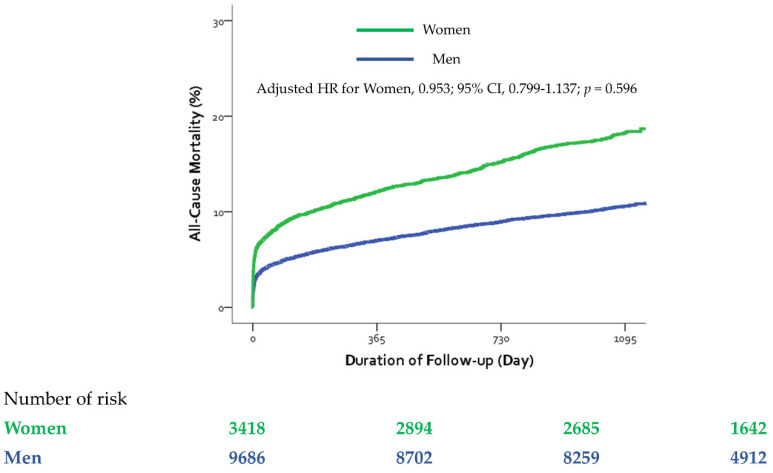
Cumulative incidence of all-cause mortality in the entire sample. CI = confidence interval, HR = hazard ratio.

**Figure 2 jcm-11-00541-f002:**
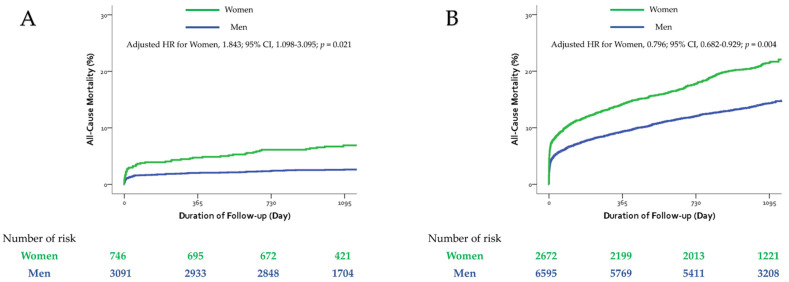
Cumulative incidence of all-cause mortality. (**A**) In the young age group, and (**B**) in the elderly age group. CI = confidence interval, HR = hazard ratio.

**Table 1 jcm-11-00541-t001:** Demographic and clinical characteristics in women and men according to age group.

	Young			Elderly			Interaction Effect *p* Value
	Women (*n* = 746)	Men (*n* = 3091)	*p* Value	Women (*n* = 2672)	Men (*n* = 6595)	*p* Value	
Age (year)	56.5 ± 6.7	47.4 ± 5.4	<0.001	76.4 ± 6.4	67.5 ± 8.5	<0.001	0.496
BMI (kg/m^2^)	24.2 ± 3.6	25.5 ± 3.3	<0.001	23.0 ± 3.6	23.7 ± 3.0	<0.001	<0.001
No chest pain	92 (12.3)	223 (7.2)	<0.001	538 (20.1)	957 (14.5)	<0.001	0.171
Hypertension	360 (48.3)	1006 (32.5)	<0.001	1912 (71.6)	3412 (51.7)	<0.001	0.044
Diabetes	237 (31.8)	557 (18.0)	<0.001	958 (35.9)	2000 (30.3)	<0.001	<0.001
Dyslipidemia	104 (13.9)	405 (13.1)	0.545	268 (10.0)	697 (10.6)	0.442	0.355
Previous myocardial infarction	45 (6.0)	177 (5.7)	0.748	205 (7.7)	602 (9.1)	0.024	0.201
Previous cerebrovascular accident	44 (5.9)	70 (2.3)	<0.001	270 (10.1)	504 (7.6)	<0.001	0.001
Current smoking	95 (12.7)	2248 (72.7)	<0.001	162 (6.1)	2608 (39.5)	<0.001	<0.001
SBP (mmHg)	131.5 ± 30.3	132.5 ± 29.9	0.422	129.2 ± 31.4	129.3 ± 29.5	0.939	0.509
DBP (mmHg)	78.7 ± 18.3	82.3 ± 19.2	<0.001	76.5 ± 18.5	77.8 ± 17.6	0.002	0.007
HR (/min)	79.2 ± 18.0	79.0 ± 18.3	0.780	80.3 ± 21.0	77.8 ± 19.7	<0.001	0.012
Killip class			<0.001			<0.001	0.334
I	598 (80.2)	2704 (87.5)		1816 (68.0)	5102 (77.4)		
II	60 (8.0)	157 (5.1)		343 (12.8)	573 (8.7)		
III	57 (7.6)	86 (2.8)		331 (12.4)	503 (7.6)		
IV	31 (4.2)	144 (4.7)		181 (6.8)	417 (6.3)		
Killip class ≥ II	148 (19.8)	387 (12.5)	<0.001	855 (32.0)	1493 (22.6)	<0.001	0.542
WBC (10^3^/μL)	9928 ± 4116	11,549 ± 4054	<0.001	10,146 ± 4354	10,230 ± 4818	0.433	<0.001
Hemoglobin (g/dL)	12.8 ± 1.8	15.2 ± 1.5	<0.001	12.0 ± 1.7	13.9 ± 2.0	<0.001	<0.001
Anaemia	183 (24.5)	159 (5.1)	<0.001	1186 (44.4)	1700 (25.8)	<0.001	<0.001
Glucose (mg/dL)	184.6 ± 107.7	161.1 ± 73.4	<0.001	180.8 ± 93.1	168.1 ± 77.9	<0.001	0.006
eCCr (mL/min/1.73 m^2^)	90.2 ± 43.9	94.5 ± 35.8	0.013	72.7 ± 42.2	79.2 ± 36.4	<0.001	0.201
Renal insufficiency	145 (19.4)	257 (8.3)	<0.001	1003 (37.5)	1617 (24.5)	<0.001	0.003
Peak CK-MB (ng/mL)	83.4 ± 117.9	133.6 ± 174.9	<0.001	89.0 ± 140.3	111.6 ± 171.3	<0.001	<0.001
LDL-C (mg/dL)	116.3 ± 41.9	123.4 ± 40.3	<0.001	109.9 ± 42.4	106.5 ± 38.6	0.001	<0.001
STEMI	316 (42.4)	1757 (56.8)	<0.001	1094 (40.9)	3158 (47.9)	<0.001	0.001
LVEF (%)	53.5 ± 11.8	53.5 ± 10.1	0.981	50.8 ± 11.8	51.4 ± 11.3	0.565	0.302

BMI = body mass index, CK-MB = creatine kinase-myocardial, DBP = diastolic blood pressure, eCCr = estimated creatinine clearance rate, HR = heart rate, LDL-C = low density lipoprotein cholesterol, LVEF = left ventricular ejection fraction, SBP = systolic blood pressure, STEMI = ST segment elevation myocardial infarction, WBC = white blood cell count.

**Table 2 jcm-11-00541-t002:** In-hospital characteristics in women and men according to age group.

	Young			Elderly			Interaction Effect *p* Value
	Women (*n* = 746)	Men (*n* = 3091)	*p* Value	Women (*n* = 2672)	Men (*n* = 6595)	*p* Value	
Coronary angiography	740 (99.2)	3083 (99.7)	0.027	2580 (96.6)	6492 (98.4)	<0.001	0.557
GPIIbIIIa	77 (10.3)	569 (18.4)	<0.001	273 (10.2)	891 (13.5)	<0.001	0.016
IVUS	134 (18.0)	601 (19.4)	0.356	350 (13.1)	1248 (18.9)	<0.001	0.006
Extent of coronary disease			<0.001			<0.001	0.118
Non obstructive	76 (10.3)	99 (3.2)		129 (5.0)	204 (3.1)		
1 vessel disease	369 (49.9)	1743 (565)		1052 (40.8)	2804 (43.2)		
2 vessel disease	183 (24.7)	802 (26.0)		739 (28.6)	1888 (29.1)		
3 vessel disease	112 (15.1)	439 (14.2)		660 (25.6)	1596 (24.6)		
MINOCA	76 (10.3)	99 (3.2)	<0.001	129 (5.0)	204 (3.1)	<0.001	<0.001
MVD	295 (39.9)	1241 (40.3)	0.847	1399 (54.2)	3438 (53.7)	0.630	0.686
Culprit			0.075			0.018	0.016
LAD	333 (53.2)	1414 (49.0)		1072 (46.7)	2657 (44.7)		
LCx	96 (15.3)	507 (17.6)		424 (18.5)	1026 (17.3)		
RCA	184 (29.4)	930 (32.2)		752 (32.8)	2085 (35.1)		
LM	13 (2.1)	37 (1.3)		48 (2.1)	176 (3.0)		
Anterior AMI	346 (55.3)	1451 (50.2)	0.022	1120 (48.8)	2833 (47.7)	0.362	0.121
Pre-PCI TIMI flow ≥ 2	283 (45.2)	1029 (35.6)	<0.001	1021 (44.5)	2617 (44.0)	0.718	<0.001
PCI	622 (84.1)	2879 (93.4)	<0.001	2287 (88.6)	5920 (91.2)	<0.001	<0.001
Stenting	571 (77.2)	2700 (87.6)	<0.001	2088 (80.9)	5517 (85.0)	<0.001	<0.001
MV PCI	148 (20.0)	580 (18.8)	0.460	629 (24.4()	1582 (24.4)	0.991	0.517
Successful PCI	619 (99.4)	2853 (98.9)	0.503	2245 (97.9)	5863 (98.8)	0.003	0.056
Complete revascularization	468 (75.1)	2165 (75.1)	0.979	1470 (64.1)	3960 (66.7)	0.025	0.302
Major bleeding	12 (1.6)	50 (1.6)	0.986	64 (2.4)	141 (2.1)	0.446	0.734
Mechanical cardiac support	20 (2.7)	77 (2.5)	0.767	116 (4.3)	289 (4.4)	0.931	0.759
Aspirin	742 (99.5)	3079 (99.6)	0.532	2655 (99.4)	6557 (99.4)	0.733	0.729
P2Y12 inhibitors	737 (98.8)	3074 (99.5)	0.05	2647 (99.1)	6525 (98.9)	0.586	0.053
Potent P2Y12 inhibitors	249 (33.4)	1315 (42.5)	<0.001	606 (22.7)	2188 (33.2)	<0.001	0.179
Beta-blockers	596 (79.9)	2651 (85.8)	<0.001	2094 (78.4)	5254 (79.7)	0.162	0.004
RAAS blocker	563 (75.5)	2478 (80.2)	0.004	2013 (75.3)	4996 (75.8)	0.671	0.023
Statin	673 (90.2)	2932 (94.9)	<0.001	2321 (86.9)	5914 (89.7)	<0.001	0.010

AMI = acute myocardial infarction, GPIIbIIIa = glycoprotein IIb/IIIa inhibitors, IVUS = intravascular ultrasound, LAD = left anterior descending, LCx = left circumflex, MINOCA = myocardial infarction with nonobstructive coronary arteries, MV = multi-vessel, MVD = multi-vessel disease, PCI = percutaneous coronary interventions, RAAS = renin–angiotensin–aldosterone system, RCA = right coronary artery, TIMI = thrombolysis in myocardial infarction.

**Table 3 jcm-11-00541-t003:** Relationship between gender and clinical outcomes after acute myocardial infarction in the entire cohort.

	Women (*n* = 3418)	Men (*n* = 9686)	Adjusted HR for Women	95% CI	*p* Value
In-Hospital Mortality	182 (5.3)	322 (3.3)	1.752	1.060–2.895	0.029
At 3-Year					
All-Cause Mortality	607 (17.8)	1002 (10.3)	0.953	0.799–1.137	0.596
Cardiac Mortality	386 (11.3)	607 (6.3)	0.888	0.722–1.093	0.263
re-MI	143 (4.2)	333 (3.4)	0.981	0.758–1.269	0.882
re-HHF	233 (6.8)	278 (2.9)	1.225	0.979–1.533	0.076
MACE	873 (25.5)	1447 (14.9)	0.930	0.815–1.063	0.288

MACE (Major Adverse Cardiac Events) = a composite of all-cause mortality, recurrent myocardial infarction (re-MI), and re-hospitalisation for heart failure (re-HHF) at 3 years. CI = confidence interval, HR = hazard ratio.

**Table 4 jcm-11-00541-t004:** Relationship between gender and clinical outcomes after acute myocardial infarction for two age categories.

	Young (*n* = 3837)					Elderly (*n* = 9267)					*p* Value for Interaction
	Women (*n* = 746)	Men (*n* = 3091)	Adjusted HR for Women	95% CI	*p* Value	Women (*n* = 2672)	Men (*n* = 6595)	Adjusted HR for Women	95% CI	*p* Value	
In-Hospital Mortality	20 (2.7)	36 (1.2)	3.305	1.211–9.019	0.020	162 (6.1)	286 (4.3)	1.199	0.811–1.774	0.362	0.020
at 3-Year											
All-Cause Mortality	51 (6.8)	79 (2.6)	1.843	1.098–3.095	0.021	556 (20.8)	923 (14.0)	0.796	0.682–0.929	0.004	0.003
Cardiac Mortality	30 (4.0)	58 (1.9)	0.851	0.275–2.636	0.779	356 (13.3)	549 (8.3)	0.945	0.758–1.177	0.613	0.255
re-MI	32 (4.3)	86 (2.8)	1.024	0.613–1.711	0.926	111 (4.2)	247 (3.7)	1.081	0.844–1.384	0.538	0.202
re-HHF	20 (2.7)	26 (0.8)	1.328	0.513–3.436	0.558	213 (8.0)	252 (3.8)	1.735	1.388–2.170	<0.001	0.226
MACE	92 (12.3)	181 (5.9)	1.398	0.992–1.970	0.056	781 (29.2)	1266 (19.2)	0.902	0.787–1.033	0.137	0.037

MACE (Major Adverse Cardiac Events) = a composite of all-cause mortality, recurrent myocardial infarction (re-MI), and re-hospitalisation for heart failure (re-HHF) at 3 years. CI = confidence interval, HR = hazard ratio.

## Data Availability

The data underlying this article were provided by the Korea Acute Myocardial Infarction Registry–National Institutes of Health committee by permission. Data will be shared on request to the corresponding author with the permission of the Korea Acute Myocardial Infarction Registry–National Institutes of Health committee.

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
