# Peer review of "Gender Differences in All-Cause Mortality after Acute Myocardial Infarction: Evidence for a Gender–Age Interaction"

_jcm, 2022, doi:10.3390/jcm11030541_

Round 1

Reviewer 1 Report

Pil Sang Song, Jin-Ok Jeong and colleagues in their paper report the interesting findings from a retrospective analysis focused on the role of sex and age on the mortality risk after acute myocardial infarction. The analysis was performed on a large sample size from the Korea Acute Myocardial Infarction Registry-National Institute of Health during a relatively short enrolment period. They found that age and sex significantly influence the risk of death after acute myocardial infarction. The paper is methodologically sound and quite well written.

I have the following concerns:

Main points

  • The decision to cathegorise patients in young if they were <65 years (men)/<55 years (women) appears quite arbitrary, especially in consideration of the average age of AMI in women (72 years old). This decision should be better discussed or other analyses considering different cut-offs should be reported.
  • The main analysis considered an all-cause mortality endpoint. However, the cardiac death did not show a significant higher incidence in women compared to men. In this view, may the all-cause death be due to other causes? Please discuss.
  • Women appear to be treated less frequently with antineurohormonal therapy. However, in heart failure setting it was shown that female sex is a protective factor for adverse outcome (Cannata, A. et al. Sex-Specific Prognostic Implications in Dilated Cardiomyopathy After Left Ventricular Reverse Remodeling. J. Clin. Med. 2020, 9, 2426. https://doi.org/10.3390/jcm9082426). May it be hypothesized that women benefit less than men from antineurohormonal therapy, both in ischemic heart disease and in heart failure? Please discuss.
  • The variables considered for the adjustment in the multivariable analysis are not reported. This is crucial, as the variables to be considered should be derived from univariable analysis.

Minor points

  • I would suggest to move the aims of the study at the end of the abstract.
  • I agree that 95% C.I. are more informative than p value, but I would suggest to add the p value in the text.
  • In the abstract it is not clear enough the comparison perfromed (young women vs young men or young women vs men irrespectively from age)?
  • All the patients received anti P2Y12 before PCI. However, Prasugrel is currently contraindicated in patients with unknown coronary artery anatomy. This point should be addressed.
  • The secondary endpoint should be briefly reported in the abstract
  • It is interesting to notice that the 3-years risk of death is lower in women compared to men if we look at the elderly. Is it possible that amongst elderly the role of comorbidities is balanced between men and women?
  • The lack of a validation cohort should be considered as a limitation.

Author Response

Main points

  1. The decision to cathegorise patients in young if they were <65 years (men)/<55 years (women) appears quite arbitrary, especially in consideration of the average age of AMI in women (72 years old). This decision should be better discussed or other analyses considering different cut-offs should be reported.

Response

Thank you very much for your comment. We absolutely agree with the reviewer’s concern about these problems. As mentioned in the METHODS section of the original manuscript, for the current analysis, we defined “young” as <55 years of age in men and as <65 years of age in women. This age cutoff point was chosen because women are often older when they present with their first AMI, at an average age of 71.8 years compared with 65 years for men [Reference 1]. However, creating this artificial definition of different age cut-offs can be quite arbitrary and problematic, especially when examining an age-gender interaction. To respond to the reviewer’s recommendation, we performed different analyzes taking different cut-offs into account. In this sensitivity analysis, we defined “young” as ≤45 years of age in men and ≤55 years of age in women. Although the gender-age interaction between men and women was not statistically significant (P value for the interaction between gender and age was 0.626), hazards ratios calculated from this model indicated that the hazards of 3-years all-cause mortality were almost 263% higher in women compared with men in the age group of young (adjusted HR, 2.632, 95% CI, 1.164-5.952, P = 0.020), while the hazards of 3-years all-cause mortality were about 13% lower among women compared with men in the age group of elderly (adjusted HR, 0.871, 95% CI, 0.751-1.010, P = 0.068). This sensitivity analysis indicated that the younger the age group, the worse the clinical outcomes of women compared with men. Taking into account the concerns of the reviewer, we added the following sentence in Study Limitations of the revised manuscript. “Our artificial definition of different age cut-offs between genders can be quite arbitrary, especially when examining a gender-age interaction.

Reference 1.

Arnett DK, Blumenthal RS, Albert MA, Buroker AB, Goldberger ZD, Hahn EJ, et al. 2019 ACC/AHA Guideline on the Primary Prevention of Cardiovascular Disease: Executive Summary: A Report of the American College of Cardiology/American Heart Association Task Force on Clinical Practice Guidelines. Circulation 2019;140:e563–e595.

  1. The main analysis considered an all-cause mortality endpoint. However, the cardiac death did not show a significant higher incidence in women compared to men. In this view, may the all-cause death be due to other causes? Please discuss.

Response

We appreciate this comment. This is an excellent point of view. In the entire cohort from our registry, cardiac death did not show a significantly higher incidence in women compared to men. In this view, the all-cause death may be due to other causes (non-cardiac death). However, cause of death is difficult to ascertain. Cause determination is, therefore, sometimes undertaken inaccurately in national registry studies of percutaneous coronary intervention or acute myocardial infarction. Claire E. et al. previously assessed cause-specific long-term mortality after index percutaneous coronary intervention [Reference 1]. They reported that the commonest causes of mortality were chronic non-cardiac deaths in women (5-year cumulative mortality, 5.4% but cancer and myocardial infarction/sudden death in men (5.4% and 4.3%). This is accounted for by baseline age and comorbidities rather than an additional sex-specific factor. These observations are consistent with our study of large AMI populations. Women had a higher mortality rate from all causes than men in the younger group, and these excess mortalities were presumed to be due to non-cardiac death. Following the reviewer’s recommendation, we added the following paragraph to Study Limitations of the revised manuscript. “Follow-up data were collected prospectively, and cause of death was typically determined through prospective scheduled surveillance in the KAMIR-NIH registry. Despite this methodology, however, it is possible that misclassification of causes of death occurred in some. More so, the risk of misclassification is expected to be greater in populations with multiple competing comorbidities, such as women in the young group of our study cohort.”

Reference 1

Raphael CE, Singh M, Bell M, Crusan D, Lennon RJ, Lerman A, et al. Sex Differences in Long-Term Cause-Specific Mortality After Percutaneous Coronary Intervention: Temporal Trends and Mechanisms. Circ Cardiovasc Interv 2018;11:e006062.

  1. Women appear to be treated less frequently with antineurohormonal therapy. However, in heart failure setting it was shown that female sex is a protective factor for adverse outcome (Cannata, A. et al. Sex-Specific Prognostic Implications in Dilated Cardiomyopathy After Left Ventricular Reverse Remodeling. J. Clin. Med. 2020, 9, 2426. https://doi.org/10.3390/jcm9082426). May it be hypothesized that women benefit less than men from antineurohormonal therapy, both in ischemic heart disease and in heart failure? Please discuss.

Response

We appreciate this comment. But we can't pinpoint your concerns. In the young age group, women appear to be treated less frequently with anti-neurohormonal therapies, such as beta-blockers or RAAS blockers. Even after adjustments of these confounding factors, however, the risks of all-cause mortality were almost 84% higher in women compared with men in the young age group (adjusted HR, 1.843; 95% CI, 1.098-3.095, P = 0.021). In this multiple adjustment model, the risk factors of all-cause mortality were gender (for women adjusted HR 1.928, 95% CI 1.119-3.322), chest pain at presentation (adjusted HR 0.331, 95% CI 0.184-0.594), prior myocardial infarction (adjusted HR 2.628, 95% CI 1.344-5.140), initial hemoglobin (adjusted HR 0.857, 95% CI 0.771-0.953), in-hospital left ventricular ejection fraction on echocardiography (adjusted HR 0.951, 95% CI 0.930-0.972), major bleeding during index hospitalization (adjusted HR 7.070, 95% CI 3.308-15.110), mechanical circulatory support during index hospitalization (adjusted HR 3.978, 95% CI 1.938-8.162), and beta-blocker prescription (adjusted HR 0.468, 95% CI 0.262-0.839).

As you mentioned, it has been shown that the female sex is a protective factor for adverse outcomes in heart failure settings. Cannata, A. et al. reported that females achieving left ventricular reverse remodeling experienced a more favourable long-term prognosis and male sex has been confirmed as independently associated with adverse prognosis even after the left ventricular reverse remodeling is achieved. To test the reviewer's hypothesis, we conducted an additional analysis. This part of the analysis assessed the role of the gender-antineurohormonal therapies interaction in the young age group and provided separate estimates of the association between anti-neurohormonal therapies and all-cause mortality in women and men. There was no significant interaction between gender and anti-neurohormonal therapies for all-cause mortality in the young age group of our AMI cohort (interaction P value 0.458 between gender and RAAS blockers, and interaction P value 0.111 between gender and beta-blocker blockers). We did not add this analysis to the revised manuscript because we thought it did not fit the main topic of our paper.

  1. The variables considered for the adjustment in the multivariable analysis are not reported. This is crucial, as the variables to be considered should be derived from univariable analysis.

Response

Thanks for your comments. However, in the METHOD, Statistical Analysis section of the original manuscript, multivariable analysis was described as follows, “In the third model, other characteristics were added. These added characteristics were baseline variables with P < 0.05 in the univariate analysis and any other baseline variables judged to be of clinical relevance from previously published literature: age categorised into 2 age groups (young versus elderly); gender; body mass index; no chest pain at presentation; Killip class; systolic blood pressure; heart rate; current smoker status; diabetes mellitus; hypertension; dyslipidemia; previous myocardial infarction; previous cerebrovascular accident; haemoglobin; white blood cell count; creatinine clearance; left ventricular ejection fraction; coronary angiography; use of glycoprotein IIb/IIIa inhibitors; multivessel disease; anterior AMI; post-PCI TIMI (thrombolysis in myocardial infarction) 2 or 3 flow; major bleeding or acute kidney injury during hospitalisation; mechanical cardiac support; and use of beta-blockers, RAAS blockers, potent P2Y12 inhibitors (ticagrelor or prasugrel), or oral anticoagulants at discharge.” Please check again.

Minor points

  1. I would suggest to move the aims of the study at the end of the abstract.

Response

We appreciate this comment. Following the reviewer’s recommendation, the authors changed the ABSTRACT in the revised manuscript to the following;

Studies of gender differences in mortality after acute myocardial infarction (AMI) have shown conflicting results.” A total of 13,104 patients from the KAMIR-NIH between November 2011 and December 2015, was classified into young (n = 3,837 [29.3%]) and elderly (n = 9,267 [70.7%]) patients. For the study, women <65 and men <55 years of age were considered “young”. In the adjusted model of the entire cohort, there was no significant difference in three-year all-cause mortality between women and men (17.8% vs. 10.3%; adjusted hazard ratio [HR], 0.953; 95% confidence interval [CI], 0.799-1.137). However, when the entire cohort was subdivided into two age groups, young women showed an 84.3% higher mortality rate than men (adjusted HR, 1.843; 95% CI, 1.098-3.095). Contrariwise, elderly women patients had a 20.4% lower hazard of mortality compared with men (adjusted HR, 0.796; 95% CI, 0.682-0.929). The interaction of gender with age was significant, even after multiple adjustments (adjusted P for interaction =0.003). “The purpose of this study was to assess whether gender differences depend on the patients' age. Based on our analysis,” the higher mortality of young women remains even in the contemporary era of AMI. A better understanding of the mechanisms underlying these differences is warranted.

  1. I agree that 95% C.I. are more informative than p value, but I would suggest to add the p value in the text.

Response

We agree with the reviewer’s opinion. Owing to the limitation of word counts, however, we could not provide P value in the ABSTRACT section of the revised manuscript. Instead, all P values are presented in other parts (main text) of the revised manuscript.

  1. In the abstract it is not clear enough the comparison performed (young women vs young men or young women vs men irrespectively from age)?

Response

We are sorry that an unclear description might cause confusion. In the revised manuscript, we changed as follows; “However, when the entire cohort was subdivided into two age groups, young women showed an 84.3% higher mortality rate than young men (adjusted HR, 1.843; 95% CI, 1.098-3.095). Contrariwise, elderly women patients had a 20.4% lower hazard of mortality compared with elderly men (adjusted HR, 0.796; 95% CI, 0.682-0.929).”

  1. All the patients received anti P2Y12 before PCI. However, Prasugrel is currently contraindicated in patients with unknown coronary artery anatomy. This point should be addressed.

Response

We appreciate this comment. This is an excellent point of view. We agree with the reviewer’s concern about prasugrel. According to the recent ESC NSTE-AMI guidelines [Reference 1], the loading of prasugrel is currently not recommended in patients with NSTE-AMI before performing CAG. However, the KAMIR-NIH registry was implemented from November 2011 to December 2015. At that time, in Korea, it is generally recommended that all patients with suspected NSTE-AMI receive a loading dose of aspirin (300 mg) and a P2Y 12 inhibitor (ticagrelor 180 mg; prasugrel 60 mg; or clopidogrel 300 to 600 mg) prior to the procedure.

Reference 1

Collet JP, Thiele H, Barbato E, Barthélémy O, Bauersachs J, Bhatt DL, et al. 2020 ESC Guidelines for the management of acute coronary syndromes in patients presenting without persistent ST-segment elevation. Eur Heart J 2021;42:1289-1367. doi: 10.1093/eurheartj/ehaa575.

  1. The secondary endpoint should be briefly reported in the abstract

Response

Thank you very much for your comment. Owing to the limitation of word counts, however, we could not describe the secondary endpoint in the ABSTRACT section of the revised manuscript. Instead, in the METHODS, Study End Points and Definitions section of the revised main text, the secondary endpoints are described as follows; “The primary endpoint of the study was 3-year all-cause mortality; the secondary outcomes were in-hospital all-cause mortality, recurrent myocardial infarction, re-hospitalisation for heart failure, and major adverse cardiac events (MACE) (a composite of all-cause mortality, recurrent myocardial infarction, and re-hospitalisation for heart failure) at three years.”

  1. It is interesting to notice that the 3-years risk of death is lower in women compared to men if we look at the elderly. Is it possible that amongst elderly the role of comorbidities is balanced between men and women?

Response

Thank you very much for your comment. We absolutely agree with the reviewer’s concern about these results. However, it may be unreasonable to estimate that the 3-year risk of death in the elderly is lower in women than in men because the roles of comorbidities are balanced between men and women. Because, in the present study, we found higher rates of comorbidity and risk factors among women compared with men in both age groups, even though these adverse cardiovascular risk characteristics were less pronounced in the elderly than in young age groups. Although the reasons for the differences in long-term outcomes between men and women from the elderly population are unclear, it is possible that once the elderly women are identified as having coronary artery disease (ie, after acute myocardial infarction), they may receive increased aggressive secondary preventive care (there were no gender differences in the prescription of aspirin, P2Y12 inhibitors, beta-blockers, and RAAS blockers in our data as shown in Table 2 of the original manuscript) and attention to recurrent symptomatic coronary artery disease (as evidenced by a lower rate of any repeat revascularization; unadjusted HR for any repeat revascularization of the elderly women,0.747; 95% CI, 0.632-0.882; P = 0.001. This result was not presented in the original manuscript), thereby resulting in better long-term clinical outcomes than those of the elderly men in the contemporary acute myocardial infarction era.

  1. The lack of a validation cohort should be considered as a limitation.

Response

We appreciate this comment. This is an excellent point of view. We agree with the reviewer’s concern about the lack of a validation cohort. Our study addressed the east Asian AMI population alone. East Asian patients had a higher bleeding risk but a relatively low ischemic risk compared to Western patients. These may have resulted in selection bias in the population studied. Following the reviewer’s recommendation, the following sentence was added to the Study Limitations of the revised manuscript as follows; “Finally, the results of our analysis should be tested in other validation cohorts, especially Western patients.

We thank the reviewer for valuable comments. Addressing them fully has significantly strengthened the revised manuscript.

Reviewer 2 Report

I found the manuscript entitled: 'Gender Differences in All-Cause Mortality after Acute Myocardial Infarction: Evidence for a Gender-Age Interaction' very interesting. The study conducted by the authors is of great clinical relevance. They are well-documented methodologically, and the obtained results are exhaustively discussed in the discussion. The authors have also properly presented the limitations of the study.

Author Response

We thank the reviewer for valuable comments.

Reviewer 3 Report

The results of this study bring novelty regarding the interaction between age and sex in patients with myocardial infarction, and the authors have succeeded to emphasize the need to raise awareness of cardiovascular risk in young women.

My only suggestion is to add more references next to reference number 1 (page 1, line 35)

Author Response

Response

We appreciate this comment. Following the reviewer’s recommendation, the authors added one more reference next to reference number 1 in the revised manuscript as follows; “Previous studies have consistently reported that women patients with acute myocardial infarction (AMI) have higher unadjusted mortality rates than men, especially for short-term follow-up [1,2].”

REFERENCES

  1. Sielski J, Kaziród-Wolski K, Jurys K, Wałek P, Siudak Z. The Effect of Periprocedural Clinical Factors Related to the Course of STEMI in Men and Women Based on the National Registry of Invasive Cardiology Procedures (ORPKI) between 2014 and 2019. J Clin Med 2021;10:5716. doi: 10.3390/jcm10235716.
  2. Mehta LS, Beckie TM, DeVon HA, Grines CL, Krumholz HM, Johnson MN, et al. Acute Myocardial Infarction in Women: A Scientific Statement From the American Heart Association. Circulation 2016;133:916-47. doi: 10.1161/CIR.0000000000000351.

We thank the reviewer for valuable comments. Addressing them fully has significantly strengthened the revised manuscript.